# Diagnostic accuracy of multiorgan point-of-care ultrasound compared with pulmonary computed tomographic angiogram in critically ill patients with suspected pulmonary embolism

Adriana M. Girardi[1,2]*, Eduardo E. Turra[2], Melina Loreto[2], Regis Albuquerque[2], Tiago S. Garcia[3,4], Tatiana H. Rech[2,4], Marcelo B. Gazzana[1,5]

**1** Postgraduate Program in Pneumological Sciences, Universidade Federal do Rio Grande do Sul, Porto Alegre, RS, Brazil, **2** Intensive Care Unit, Hospital de Clínicas de Porto Alegre, Porto Alegre, RS, Brazil, **3** Radiology Division, Hospital de Clínicas de Porto Alegre, Porto Alegre, RS, Brazil, **4** School of Medicine, Universidade Federal do Rio Grande do Sul, Porto Alegre, RS, Brazil, **5** Pulmonary Division, Hospital de Clínicas de Porto Alegre, Porto Alegre, RS, Brazil

* adriana.m.girardi@gmail.com

## Abstract

### Background

Critically ill patients have a higher incidence of pulmonary embolism (PE) than non-critically ill patients, yet no diagnostic algorithm has been validated in this population, leading to the overuse of pulmonary artery computed tomographic angiogram (CTA). This study aimed to comparatively evaluate the diagnostic accuracy of point-of-care ultrasound (POCUS) combined with laboratory data versus CTA in predicting PE in critically ill patients.

### Methods

A prospective diagnostic accuracy study. Critically ill patients with suspected acute PE undergoing CTA were prospectively enrolled. Demographic and clinical data were collected from electronic medical records. Blood samples were collected, and the Wells and revised Geneva scores were calculated. Standardized multiorgan POCUS and CTA were performed. The discriminatory power of multiorgan POCUS combined with biochemical markers was tested using ROC curves, and multivariate analysis was performed.

### Results

A total of 88 patients were included, and 37 (42%) had PE. Multivariate analysis showed a relative risk (RR) of PE of 2.79 (95% CI, 1.61–4.84) for the presence of right ventricular (RV) dysfunction, of 2.54 (95% CI, 0.89–7.20) for D-dimer levels >1000 ng/mL, and of 1.69 (95% CI, 1.12–2.63) for the absence of an alternative diagnosis to PE on lung POCUS or chest radiograph. The combination with the highest diagnostic accuracy for PE included the following variables: 1– POCUS transthoracic echocardiography with evidence of RV

**Data Availability Statement:** All relevant data are within the paper and its Supporting Information files.

**Funding:** This paper received a financial incentive to cover for publication fees by Fundo de Incentivo à Pesquisa e Eventos (FIPE) of Hospital de Clínicas de Porto Alegre (project number 2018-0282). The funders had no role in study design, data collection and analysis, decision to publish, or preparation of the manuscript.

**Competing interests:** The authors have declared that no competing interests exist.

dysfunction; 2– lung POCUS or chest radiograph without an alternative diagnosis to PE; and 3– plasma D-dimer levels >1000 ng/mL. Combining these three findings resulted in an area under the curve of 0.85 (95% CI, 0.77–0.94), with 50% sensitivity and 96% specificity.

## Conclusions

Multiorgan POCUS combined with laboratory data has acceptable diagnostic accuracy for PE compared with CTA. The combined use of these methods might reduce CTA overuse in critically ill patients.

## Introduction

Acute pulmonary embolism (PE) is the sudden occlusion of the pulmonary artery or its branches [1], and patients admitted to the intensive care unit (ICU) are at increased risk of PE [2, 3]. Diagnostic algorithms and prediction scores, such as the Wells and revised Geneva scores, are available to guide the diagnostic approach to PE in outpatient settings and are occasionally used in the ICU [4]. However, critical illness makes it difficult to diagnose PE [5], and these prediction scores remain unacceptably inaccurate in critically ill patients [6].

The pulmonary computed tomographic angiogram (CTA) is the gold standard for diagnosing PE [7] and has been widely used in critically ill patients. However, the potential overuse of CTA unnecessarily exposes patients to ionizing radiation, iodinated contrast media, and transfer risks. In addition, many patients with PE have severe hypoxemia and hemodynamic instability that precludes CTA use [8], requiring alternative methods for diagnosis.

The use of point-of-care ultrasound (POCUS) in the ICU has emerged as an excellent diagnostic tool [9–11]. The investigation of PE with multiorgan ultrasound (US), involving cardiac, pulmonary, and lower limb venous scans, has shown promising results in emergency departments [12, 13], suggesting its potential role in PE detection in critically ill patients.

PE is associated with high mortality and has a nonspecific clinical presentation. Therefore, alternative diagnostic strategies to help with timely decisions are important in critically ill patients. The purpose of this study was to comparatively evaluate the diagnostic accuracy of multiorgan POCUS combined with laboratory data versus CTA in predicting PE in critically ill patients.

## Materials and methods

This prospective diagnostic accuracy study is reported according to the Standards for Reporting of Diagnostic Accuracy Studies (STARD) guidelines [14] and involved comparing the results of the index test (multiorgan POCUS combined with laboratory data) and reference standard (CTA), defined as the best available method for establishing the presence or absence of the condition of interest (PE) (S1 File).

All study procedures followed the tenets of the Declaration of Helsinki of 1975 and were performed in accordance with the guidelines of the Research Ethics Committee of the Hospital de Clínicas de Porto Alegre, Brazil, for research involving human subjects. Our study was reviewed and approved by the Research Ethics Committee of the Hospital de Clínicas de Porto Alegre, Brazil, in September 2018 (number 2018–0282). Written informed consent was obtained from each study participant or legal representative, and the patient was only included in the study after this consent.

The patients were enrolled from September 2018 to February 2020. Critically ill adult patients with suspected acute PE for whom the attending physician ordered pulmonary artery CTA were included.

Demographic and clinical data were collected from electronic medical records. Blood samples were collected for D-dimer, troponin, and N-terminal pro-B-type natriuretic peptide (NT-proBNP) level measurements at study entry. The Wells score and the revised Geneva score were calculated by the same researcher for all patients. The study protocol included standardized multiorgan POCUS and CTA. Multiorgan POCUS involved performing lower limb deep vein US, lung US, and transthoracic echocardiography (TTE).

POCUS examinations were performed independently by two critical care fellows with advanced training in POCUS who had undergone three months of training in image acquisition specifically for the study protocol. All images were saved and reviewed by a senior physician if necessary. The researchers performing the POCUS examinations were blinded to the CTA results. POCUS examinations and CTA were performed no more than 24 hours apart.

Lung US was performed with a convex probe. An A- or B-pattern was defined on the anterior chest. The presence or absence of posterolateral alveolar pleural syndrome (PLAPS) was determined on the posterolateral chest [15]. Images of the anterosuperior and anteroinferior quadrants were acquired on the midclavicular line at the second and fifth intercostal spaces, respectively. Images of the posterolateral chest were obtained between the mid-and posterior axillary lines, with the inferior posterolateral quadrant located at the thoracoabdominal transition. The presence of one or more of the following lung abnormalities was considered an alternative diagnosis to PE on lung US: 1- the absence of pleural slip suggesting the presence of pneumothorax; 2- the presence of a hypoechoic pleural-based lesion suggesting consolidation or pulmonary atelectasis; 3- the presence of three or more B lines in an intercostal space in non-dependent lung areas suggesting alveolar-interstitial edema; 4- the presence of homogeneous anechoic area in a dependent lung area suggesting pleural effusion [15, 16].

Lower limb venous US was performed with a linear probe to scan the femoral and popliteal veins bilaterally. Three points were examined in each extremity, two in the femoral vein and one in the popliteal vein. Deep vein thrombosis (DVT) was defined as the absence of complete compression of the vessel wall with slight probe compression, with or without visualizing hyperechogenic areas within the vessel.

TTE was performed with a cardiac probe. The right ventricle (RV) to left ventricle (LV) diameter ratio (RV/LV ratio) and the tricuspid annular plane systolic excursion (TAPSE) in the apical four-chamber view were assessed. An RV/LV ratio $\geq 1.0$ or a TAPSE $< 1.7$ cm were considered abnormal [1].

The following parameters were evaluated: epidemiological and clinical data, electrocardiograms, chest radiographs, and laboratory tests, including ultrasensitive troponin I, D-dimers, and NT-proBNP. D-dimer levels were quantified by latex agglutination assay and considered elevated if $>400$ ng/mL (laboratory reference value). Ultrasensitive troponin I levels were determined by chemiluminescent microparticle immunoassay, and levels $>52$ pg/mL suggest myocardial injury (laboratory reference value). NT-proBNP levels were determined by electrochemiluminescence and considered elevated if $>125$ pg/mL (laboratory reference value).

Chest radiograph and electrocardiogram findings were classified as normal or abnormal. Chest radiograph abnormalities included consolidation, bilateral diffuse infiltrates, pleural effusion, or a combination of those. Electrocardiogram abnormalities included sinus bradycardia, atrioventricular block, sinus tachycardia, atrial fibrillation, atrial flutter, and S1Q3T3 pattern (a finding suggestive of RV overload).

All patients underwent CTA for the diagnosis of PE. CTA images were obtained with a 16-channel CT scanner in helical scan mode. The images were considered positive or negative

for PE according to the report of two independent radiologists, one of them with expertise in thoracic radiology. In case of disagreement, a third radiologist analyzed the images. In patients with PE, thrombus location was defined according to the caliber of the affected vessel and classified as follows: main artery, lobar, segmental, or subsegmental. Scans were considered negative in the presence of adequate opacification of the pulmonary artery without filling defects [17].

The sample size was calculated using PEPI software (version 11.65, 2016). The calculation was based on a previously reported maximum sensitivity of multiorgan POCUS (lower limbs, lung, and heart) of 90% for PE detection [13] and incidence of 30% of PE on CTA in critically ill patients [6]. To detect a 10% difference in the sensitivity to detect PE between the combined findings of multiorgan POCUS and CTA, with an accuracy of 12% and an alpha error of 0.05, a sample size of 85 patients was necessary.

Categorical variables were expressed as percentages. Quantitative data were expressed as mean and standard deviation if normally distributed or as median and interquartile range if not normally distributed. Interobserver agreement of CTA readings was assessed by kappa (κ) statistics, with κ values of 0.4–0.6 indicating moderate agreement, 0.61–0.8 high agreement, and 0.81–1.0 very high agreement. As appropriate, groups were compared by Student's t-test, Mann-Whitney U test, or chi-square test. Variables with $p < 0.1$ in univariate analysis and those with clinical relevance were included in multivariate analysis.

The diagnostic accuracy of POCUS and laboratory tests was analyzed using receiver operating characteristic (ROC) curves when the variables were continuous. ROC curves were also constructed by combining categorical variables after logistic regression. A $p < 0.05$ was considered statistically significant. All statistical analyses were performed using SPSS software, version 21.0 (Chicago, IL, USA).

## Results

Eighty-eight critically ill patients were included in the study. Two patients had a history of previous PE or DVT, but none of them were being treated for PE or DVT in the last six months. The mean patient age was 58 ± 15 years. Most patients were women (53%), and the mean Simplified Acute Physiology Score 3 (SAPS 3) was 60 ± 15. The main reason for ICU admission was acute respiratory failure (63%), followed by cardiopulmonary arrest (17%) and circulatory shock (12%). ICU mortality and 30-day hospital mortality were 36% and 43%, respectively.

Of 88 patients, 37 (42%) had PE detected on pulmonary CTA examination. Of these, 12 patients (32%) had PE in the main artery, three (8%) in the lobar branch, 18 (49%) in the segmental branch, and four patients (11%) in the subsegmental branch. The agreement in PE diagnosis between the two radiologists was very high (κ = 0.89).

Table 1 shows the main characteristics of the 88 patients. Patients were divided into two groups according to the presence or absence of PE. Patients without PE had a higher SAPS 3, more commonly developed chronic kidney disease and sepsis, and more often required mechanical ventilation. The Wells score was significantly higher in patients with PE than in those without PE (4.2 ± 2.5 vs. 2.9 ± 1.9; p = 0.012), but the Geneva score was not. The groups with and without PE did not differ significantly in mortality (37% vs. 47%; p = 0.5), length of ICU stay (5 [3–16] vs. 9 [5–20] days; p = 0.28), or length of hospital stay (21 [12–39] vs. 18 [13–44] days; p = 0.68).

The mean time interval between multiorgan POCUS examinations and CTA was 10 ± 7.5 hours. Of the 37 patients with positive findings for PE on CTA, 22 (60%) had RV dysfunction on POCUS TTE, with 63% sensitivity and 85% specificity. All patients were considered to have acute RV dysfunction. Twenty-eight patients (76%) had no alternative diagnosis to PE on lung

**Table 1. Characteristics of the 88 patients included in the study.**

| Characteristics | All patients (n = 88) | With PE (n = 37) | Without PE (n = 51) | P |
|---|---|---|---|---|
| **Demographics** | | | | |
| Age (years) | 58 ± 15 | 57 ± 15 | 59 ± 15 | 0.73 |
| Male gender (n,%) | 41 (47%) | 15 (40%) | 26 (50%) | 0.45 |
| SAPS 3 score | 60 ± 15 | 57 ± 16 | 62 ± 14 | 0.047 |
| **Comorbidities** | | | | |
| Hypertension (n,%) | 45 (51%) | 19 (51%) | 26 (50%) | 1 |
| Diabetes (n,%) | 24 (27%) | 7 (19%) | 17 (33%) | 0.2 |
| Chronic kidney disease (n,%) | 10 (11%) | 1 (2%) | 9 (18%) | 0.04 |
| Malignancy (n,%) | 22 (25%) | 9 (24%) | 13 (25%) | 1 |
| **Signs and symptoms** | | | | |
| Tachycardia[1] (n,%) | 44 (50%) | 20 (54%) | 24 (47%) | 0.66 |
| Tachypnea[1] (n,%) | 77 (87%) | 33 (89%) | 44 (86%) | 0.75 |
| Chest pain (n,%) | 21 (24%) | 14 (38%) | 7 (14%) | 0.018 |
| **Severity of illness** | | | | |
| Presence of sepsis (n,%) | 40 (45%) | 10 (27%) | 30 (58%) | 0.006 |
| Use of vasopressors (n,%) | 30 (34%) | 10 (32%) | 20 (39%) | 0.33 |
| Need for MV (n,%) | 55 (62%) | 18 (48%) | 37 (72%) | 0.03 |
| Need for CVC (n,%) | 57 (65%) | 20 (54%) | 37 (72%) | 0.1 |
| Immobilization[2] (n,%) | 32 (36%) | 14 (38%) | 18 (35%) | 0.9 |
| **Prediction rules** | | | | |
| Wells score | 3.4 ± 2.3 | 4.2 ± 2.5 | 2.9 ± 1.9 | 0.012 |
| Wells score >4 (n,%) | 41 (46%) | 21 (57%) | 20 (31%) | 0.15 |
| Revised Geneva score (n) | 5.5 ± 3.1 | 6.2 ± 3.4 | 5.1 ± 2.8 | 0.16 |
| Revised Geneva score >7 (n,%) | 30 (34%) | 15 (40%) | 15 (29%) | 0.39 |

CVC: central venous catheter; MV: mechanical ventilation; PE: pulmonary embolism; SAPS 3: Simplified Acute Physiology 3.

[1]Tachycardia was defined as heart rate >100 bpm and tachypnea as respiratory frequency >20 rpm.

[2]Immobilization was defined as bed restriction >48 hours.

POCUS, with 78% sensitivity and 39% specificity. Seven patients (19%) with positive findings for PE on CTA had DVT on lower limb deep vein POCUS, with 19% sensitivity and 94% specificity. The three POCUS findings (RV dysfunction, absence of an alternative diagnosis to PE on lung POCUS, and presence of lower limb DVT) occurred simultaneously in six patients (16%).

Of 88 patients, 10 (11%) had evidence of lower limb DVT on POCUS, with no significant difference between the groups with and without PE. Patients without PE tended to have an alternative pulmonary diagnosis based on lung POCUS more frequently than those with PE (35% vs. 22%; p = 0.06). Regarding POCUS TTE, patients with PE had a higher RV/LV ratio than those without PE, both in absolute values (0.98 ± 0.2 vs. 0.82 ± 0.17; p = 0.01) and when compared to the cut-off value of 1 (38% vs. 8%; p = 0.001). Patients with PE also had lower TAPSE values (1.7 ± 0.5 vs. 2.0 ± 0.4 cm; p = 0.027), indicating more significant RV dysfunction in these patients. Comparing the groups by TAPSE cut-off value above or below 1.7 cm, patients with PE had a TAPSE <1.7 cm (47% vs. 10%; p = 0.001) more frequently than those without PE (Table 2). When evaluating the performance of each POCUS examination separately, TTE with the presence of RV dysfunction showed the highest diagnostic accuracy.

**Table 2. Results of biochemical tests, imaging studies and point-of-care findings.**

| Characteristics | All patients (n = 88) | With PE (n = 37) | Without PE (n = 51) | p |
|---|---|---|---|---|
| **Biochemical and imaging results** | | | | |
| P/F ratio | 187 (118–245) | 152 (117–244) | 180 (129–220) | 0.73 |
| D-dimers (ng/mL) | 3054 (1672–4950) | 4600 (2775–5001) | 2205 (1220–3640) | 0.001 |
| N-terminal pro-BNP (pg/mL) | 1147 (269–3858) | 1420 (253–2752) | 1013 (299–4114) | 0.69 |
| Ultrasensitive troponin I (pg/mL) | 51 (27–164) | 96 (28–297) | 43 (24–90) | 0.02 |
| Abnormal electrocardiogram (n,%) | 50 (57%) | 24 (65%) | 26 (51%) | 0.14 |
| Abnormal chest X-ray (n,%) | 65 (74%) | 21 (57%) | 44 (86%) | 0.02 |
| **Point-of-care ultrasound findings** | | | | |
| Alternative diagnosis to PE on lung ultrasound (n,%) | 27 (31%) | 8 (22%) | 19 (37%) | 0.08 |
| Presence of DVT in lower limbs (n,%) | 10 (11%) | 7 (19%) | 3 (6%) | 0.08 |
| Qualitative RV dysfunction (n,%) | 9 (10%) | 7 (19%) | 2 (4%) | 0.034 |
| RV/LV ratio | 0.9 ± 0.2 | 1 ± 0.2 | 0.8 ± 0.2 | 0.01 |
| RV/LV ratio >1 (n,%) | 19 (20%) | 14 (38%) | 4 (8%) | 0.001 |
| TAPSE (cm) | 1.9 ± 0.5 | 1.7 ± 0.5 | 2.0 ± 0.4 | 0.027 |
| TAPSE <1.7cm (n,%) | 23 (26%) | 18 (47%) | 5 (10%) | 0.001 |

DVT: deep vein thrombosis; LV: left ventricle; NT-proBNP: N-terminal pro-BNP (brain natriuretic peptide); PE: pulmonary embolism; P/F ratio: ratio between partial arterial oxygen pressure and inspired oxygen fraction; RV: right ventricle; TAPSE: tricuspid annular plane systolic excursion.

Patients with PE had higher D-dimer levels than patients without PE (4600 [2775–5001] vs. 2205 [1220–3640] ng/mL; p = 0.001). All patients with D-dimer levels below the cut-off value of 400 ng/mL did not have PE on CTA.

There was no significant difference between the groups with and without PE concerning NT-proBNP levels. Patients with PE had higher, but not clinically relevant, ultrasensitive troponin I levels than patients without PE (96 [28–297] vs. 43 [24–90] pg/mL, p = 0.02).

The single laboratory parameter that improved accuracy when combined with POCUS was plasma D-dimer. Ultrasensitive troponin I and NT-proBNP did not perform well, whether alone or combined with other parameters (S2 File).

Multivariate analysis showed that the presence of RV dysfunction, assessed by the presence of RV/LV ratio ≥1 or TAPSE <1.7 cm or qualitative RV dysfunction, had a relative risk (RR) of 2.79 (95% CI, 1.61–4.84) for the presence of PE. The absence of an alternative pulmonary diagnosis, based on lung POCUS or normal chest radiograph findings, had an RR of 1.69 (95% CI, 1.12–2.63) for the presence of PE, and D-dimer levels >1000 ng/mL had an RR of 2.54 (95% CI, 0.89–7.20) for the presence of PE (Table 3).

**Table 3. Multivariate analysis of predictors of pulmonary embolism by clinical variables and multiorgan point-of-care ultrasound.**

| Variables | Odds ratio | 95% CI |
|---|---|---|
| Presence of RV dysfunction* | 2.79 | 1.61–4.84 |
| No alternative pulmonary diagnosis** | 1.69 | 1.12–2.63 |
| D-dimers >1000 ng/mL | 2.54 | 0.89–7.20 |

LV: left ventricle; RV: right ventricle; TAPSE: tricuspid annular plane systolic excursion.

95% CI: 95% confidence interval.

*RV/LV ratio >1 or TAPSE <1.7 cm or qualitative RV dysfunction.

**By lung point-of-care ultrasound or chest X-ray.

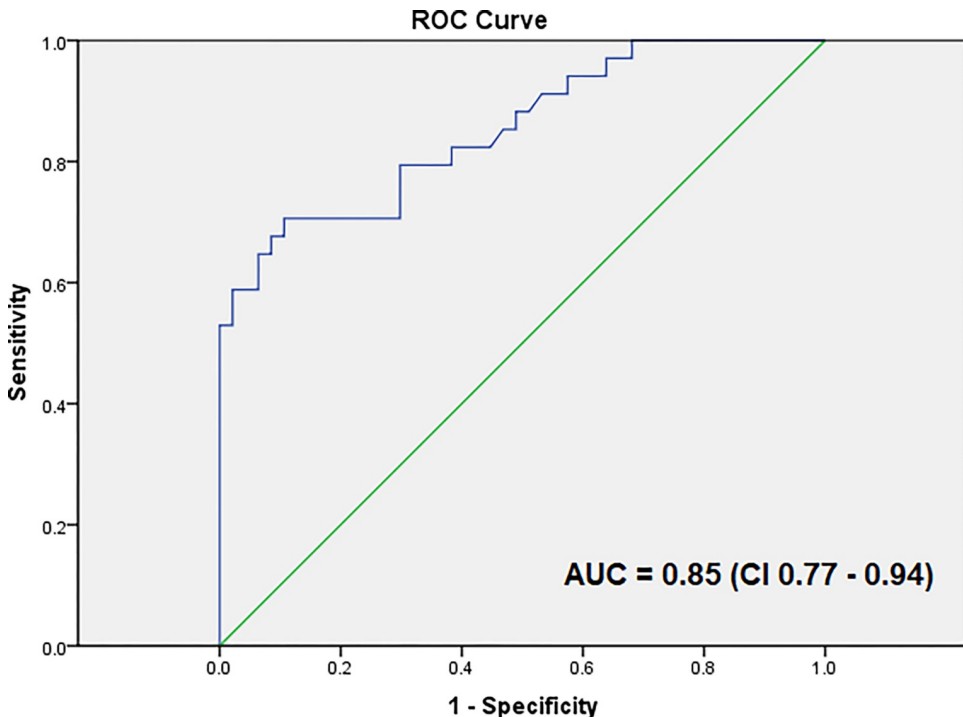

**Fig 1. ROC curves demonstrating the performance of the combination of the following three variables: 1-transthoracic echocardiography with right ventricular dysfunction; 2- lung abnormalities on lung ultrasound or chest radiograph; and 3- D-dimer levels > 1000 ng/mL in predicting pulmonary embolism.**

Combining POCUS with laboratory data provided optimal diagnostic accuracy for PE. The combination with the highest accuracy included the following three variables: 1– POCUS TTE with evidence of RV dysfunction (TAPSE <1.7 cm or RV/LV ratio ≥1 or qualitative RV dysfunction); 2– lung POCUS or chest radiograph without an alternative diagnosis to PE; and 3– plasma D-dimer levels >1000 ng/mL. Together, this combination had an area under the ROC curve (AUC) of 0.85 (95% CI, 0.77–0.94), with 50% sensitivity and 96% specificity (Fig 1).

Twenty patients (23%) presented the three positive findings concomitantly (1- presence of RV dysfunction in TTE; 2- no alternative pulmonary diagnosis; and 3- D-dimer levels >1000 ng/mL), and 18 of them had a positive PE diagnosis on CTA, with sensitivity, specificity, positive and negative predictive values of 50%, 96%, 90%, and 73%, respectively. Conversely, seven patients (8%) had the absence of the three findings concomitantly, all of them without PE on pulmonary CTA. The diagnostic accuracy of each ultrasound and combined ultrasounds for the diagnosis of PE is shown in Table 4.

## Discussion

Our findings showed that the combination of 1- POCUS TTE with evidence of RV dysfunction (TAPSE <1.7 cm or RV/LV ratio ≥1), 2- the absence of an alternative diagnosis to PE on lung POCUS or chest radiograph, and 3- D-dimer levels >1000 ng/mL has good diagnostic accuracy in predicting PE compared with CTA.

In the present study, the incidence of PE in patients undergoing pulmonary CTA for suspected PE was 42%. In non-critically ill hospitalized patients, it is estimated that 30% of CTA scans will be positive in patients with a high probability of PE [18].

**Table 4. Diagnostic accuracy of single-organ ultrasound and combined ultrasound scans for the diagnosis of pulmonary embolism.**

|  | Sens % (95% IC) | Spec % (95% IC) | PPV % (95% IC) | NPV (95% IC) | +LR (95% IC) | -LR (95% IC) |
|---|---|---|---|---|---|---|
| Cardiac ultrasound | 0.63 (0.45–0.79) | 0.85 (0.72–0.94) | 0.76 (0.56–0.90) | 0.76 (0.62–0.87) | 4.31 (2.08–2.95) | 0.43 (0.28–0.68) |
| Lung ultrasound | 0.78 (0.61–0.90) | 0.39 (0.25–0.54) | 0.48 (0.35–0.62) | 0.70 (0.50–0.86) | 1.27 (0.96–1.69) | 0.57 (0.28–1.16) |
| Limb ultrasound[1] | 0.19 (0.08–0.36) | 0.94 (0.83–0.99) | 0.70 (0.35–0.93) | 0.61 (0.49–0.72) | 3.18 (0.88–11.4) | 0.86 (0.72–1.02) |
| Multiorgan ultrasound[2] | 0.16 (0.06–0.32) | 1,00 (0.93–1.00) | 1.00 (0.54–1.00) | 0.62 (0.51–0.73) | 2.28 (0.70–10.0)[3] | 0.84 (0.73–0.97) |
| Cardiac and lung ultrasound plus D-dimers >1000ng/mL | 0.50 (0.33–0.67) | 0.96 (0.86–1.00) | 0.90 (0.68–0.99) | 0.73 (0.60–0.83) | 12.5 (3.09–50.5) | 0.52 (0.37–0.73) |

Sens: sensitivity; Spec: specificity; PPV: positive predictive value; NPV: negative predictive value; +LR: positive likelihood ratio; -LR: negative likelihood ratio

[1] Lower limb venous ultrasound

[2] Evidence of RV dysfunction and absence of an alternative pulmonary diagnosis and presence of deep venous thrombosis of the lower limbs on point-of-care ultrasound (POCUS)

[3] Calculation performed with specificity at the lower limit of 95% CI

Twenty patients had the combination of RV dysfunction, D-dimers >1000 ng/mL, and the absence of an alternative pulmonary diagnosis, and 18 (90%) of them had PE on pulmonary CTA. Furthermore, seven patients with the absence of RV dysfunction in TTE, D-dimers <1000 ng/mL, and the presence of an alternative pulmonary diagnosis had negative CTA for PE. Based on our findings, 25 patients (28%) would avoid undergoing CTA with a more comprehensive pre-test probability assessment, integrating POCUS findings and laboratory data.

TTE with signs of RV dysfunction was the component of multiorgan POCUS with the highest diagnostic accuracy. Lower limb US had the lowest impact on improving the accuracy of our results. It is estimated that 30% of patients with PE have lower limb DVT [19]. In our sample, however, 19% of patients with PE had DVT. The possibility that thrombi may originate from the upper circulation in critically ill patients, mainly associated with the presence of central venous catheters, has already been suggested in a previous study [20]. Still, this association was not found in our population of patients. Thrombi may be lodged in the iliac veins or the inferior vena cava and have already completely detached and carried into the pulmonary circulation. Our results are consistent with those of a recent study involving critically ill patients that demonstrated a lower incidence of lower limb DVT in patients with PE [21].

D-dimer testing has been used only occasionally in critically ill patients because of its low specificity in this population [22]. However, our results suggest that D-dimer levels, when combined with multiorgan POCUS, improve diagnostic accuracy for PE and may be helpful, especially if the cut-off value is adjusted for severity of disease and age [23]. In our sample, a D-dimer cut-off value >1000 ng/mL maintained excellent sensitivity, with a high negative predictive value. All patients with D-dimer levels <400 ng/mL did not have PE on pulmonary CTA, reinforcing the idea that D-dimers are useful for ruling out PE, even in critically ill patients.

Nazerian et al. [13] investigated patients with suspected PE in the emergency department and showed that multiorgan POCUS has higher diagnostic accuracy than single-organ POCUS. Our results demonstrate that the combination of laboratory data with multiorgan POCUS further improves diagnostic accuracy for PE in critically ill patients. The presence of RV dysfunction in TTE, absence of pulmonary differential diagnosis on lung POCUS or chest radiograph, and elevated D-dimer levels above 1000 ng/mL provided the best combination for the diagnosis of PE compared with CTA. Conversely, variables classically associated with PE,

such as age [23], troponin and NT-proBNP [24], and the revised Geneva score [5], did not perform well in our population, in agreement with our previous retrospective results [6]. In this prospective study, combining the Wells and Geneva prediction scores with multiorgan scans did not increase diagnostic accuracy.

In a meta-analysis published in 2021, Falster et al. [16] reported that the use of POCUS in patients with suspected PE has findings with high specificity, and its use should be encouraged to select which patients should be referred for CTA. In our study protocol, we did not assess some of the findings that have high specificity for PE, such as the McConnell's sign and the presence of pulmonary infarctions. We understand that these findings suggest PE more strongly than just RV dysfunction or absence of an alternative pulmonary diagnosis, but they are less frequent findings and require greater operator expertise for their identification. In addition, some protocols for pulmonary and cardiac assessment require specific positioning of the patient in bed, which is difficult to perform in critically ill patients with high severity.

This study has limitations. First, 11 potentially eligible patients were not included in the study due to logistic reasons. The maximum time interval of 24 hours allowed between POCUS and CTA was exceeded. Still, there is no reason to believe that these patients would differ from the included ones. Second, we defined the absence of PE on pulmonary CTA as a negative result, which is currently the preferred imaging study for diagnosing PE. However, we did not follow up the patients who survived for three months, which is considered the gold standard to exclude PE, nor did we perform autopsies on the patients who died. Third, a formal exclusion of patients with previous RV dysfunction was not performed. However, no patients were suspected to have previous pulmonary hypertension based on previous echocardiography data and clinical data. Fourth, this is a single-center study.

## Conclusion

In conclusion, multiorgan POCUS combined with laboratory data has acceptable diagnostic accuracy for PE compared with CTA and might be helpful in the ICU setting, providing an alternative method when performing a CTA is too risky and reducing unnecessary tests when the probability of disease is too high. The possibility of a 25% reduction in the need for CTA in critically ill patients with suspected PE is encouraging. Considering that multiorgan POCUS assessment of critically ill patients with suspected PE is an easy-to-perform, low-cost and low-risk technique, this diagnostic method combined with laboratory data could be easily implemented in clinical practice.

## Supporting information

**S1 File. STARD diagram.**
(PDF)

**S2 File. ROC curves demonstrate the performance of biochemical parameters in predicting pulmonary embolism.** A: Troponin. B: NT-pro-BNP.
(PDF)

**S3 File. Database with information on the 88 patients included in the final analysis.**
(XLSX)

## Acknowledgments

We thank Rogério Boff Borges for his assistance with the statistical analyses.

## Author Contributions

**Conceptualization:** Adriana M. Girardi, Tatiana H. Rech, Marcelo B. Gazzana.

**Data curation:** Adriana M. Girardi, Eduardo E. Turra.

**Formal analysis:** Adriana M. Girardi, Melina Loreto.

**Funding acquisition:** Tatiana H. Rech, Marcelo B. Gazzana.

**Investigation:** Adriana M. Girardi.

**Methodology:** Adriana M. Girardi, Regis Albuquerque, Tiago S. Garcia, Tatiana H. Rech, Marcelo B. Gazzana.

**Project administration:** Tatiana H. Rech, Marcelo B. Gazzana.

**Supervision:** Tatiana H. Rech, Marcelo B. Gazzana.

**Validation:** Tiago S. Garcia.

**Visualization:** Melina Loreto, Tatiana H. Rech.

**Writing – original draft:** Adriana M. Girardi.

**Writing – review & editing:** Tatiana H. Rech, Marcelo B. Gazzana.

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
