## [Decision Letter · Decision Letter 0]

8 Aug 2022

PONE-D-22-18103Diagnostic accuracy of multiorgan point-of-care ultrasound compared with pulmonary computed tomographic angiogram in critically ill patients with suspected pulmonary embolism.PLOS ONE

Dear Dr. Girardi, Adriana,

Thank you for submitting your manuscript to PLOS ONE. After careful consideration, we feel that it has merit but does not fully meet PLOS ONE’s publication criteria as it currently stands. Therefore, we invite you to submit a **revised version** of the manuscript that addresses the points raised during the review process.

Please submit your revised manuscript until **August 20, 2022**. If you will need more time than this to complete your revisions, please reply to this message or contact the journal office at plosone@plos.org. Please include the following items when submitting your revised manuscript:A rebuttal letter that responds to each point raised by the academic editor and reviewer(s). You should upload this letter as a separate file labeled 'Response to Reviewers'.A marked-up copy of your manuscript that highlights changes made to the original version. You should upload this as a separate file labeled 'Revised Manuscript with Track Changes'.An unmarked version of your revised paper without tracked changes. You should upload this as a separate file labeled 'Manuscript'.

We look forward to receiving your revised manuscript.

Kind regards,

Patricia Rezende do Prado

Academic Editor

PLOS ONE

**Journal Requirements:**

**Reviewer's Responses to Questions**

**Comments to the Author**

1. Is the manuscript technically sound, and do the data support the conclusions?

Reviewer #1: Yes

Reviewer #2: Partly

Reviewer #3: Partly

2. Has the statistical analysis been performed appropriately and rigorously? 

Reviewer #1: Yes

Reviewer #2: Yes

Reviewer #3: Yes

3. Have the authors made all data underlying the findings in their manuscript fully available?

Reviewer #1: Yes

Reviewer #2: Yes

Reviewer #3: No

4. Is the manuscript presented in an intelligible fashion and written in standard English?

Reviewer #1: Yes

Reviewer #2: Yes

Reviewer #3: No

5. Review Comments to the Author

Reviewer #1: • Prediction scores as Wells and Geneva for pretest probability of PE validated in outpatient settings are occasionally used in the ICU. However, none of these clinical prediction rules have been validated in the ICU population. Therefore, more researches are needed to identify features that help to accurately identify patients with PE in this setting, and to develop a suitable prediction rule for critically ill patients with suspected PE. I think that in this study I do not see the relevance of considering these scores to analyze the usefulness of the multiorgan pocus. Please, comment about it.

• In table 2, define and explain better what it means “Alternative diagnosis on lung ultrasound”.

• Lung US performed by authors was based on diagnosis of acute respiratory failure method published on 2008. I would like to suggest that you make a more current and didactic classification shown in systematic review by Falster C et al

Reviewer #2: This is an interesting study that looked into utilization of POCUS with clinical and lab values to diagnosed PE in comparison with CTPA.

"A noninvasive diagnostic strategy to help with timely decisions is important in critically ill patients" I think this statement should be modified as CTPA is already noninvasive, I would consider changing it.

Definition of altered lung result on the basis of 3 or more altered fields may miss focal lung abnormalities like consolidation or atelectasis and this definition may under-diagnose alternative diagnoses to PE.

The authors did not specify if RV dysfunction was acute or chronic as presence of previous RV dysfunction may confound the findings significantly. Also, I believe all patients with previous RV dysfunction should have been excluded from the study.

There was no mention of patients with history of previous PE or being treated for PE as these also may confound the findings.

Were the images acquired by the fellows saved and reviewed by more experienced physician to concur with accuracy of ultrasound image acquisition given that ultrasound is operator dependent?

The authors mentioned specificity of 100% in the abstract and then mentioned specificity of 96% in the results section, this should be corrected and clarified.

Given sensitivity of 50%, I think it is really difficult to state in the conclusion the diagnostic accuracy in high at this point. I would be more cautious with such a statement as the data presented does not fully support this conclusion.

Reviewer #3: I had a very difficult time determining what proportion of PEs were sub-segmental in nature. The paper also requires significant language editing and the abstract is lacking key information i.e. the sample size included in the study.

6. PLOS authors have the option to publish the peer review history of their article (what does this mean?). If published, this will include your full peer review and any attached files.

Reviewer #1: No

Reviewer #2: No

Reviewer #3: **Yes.**

---

## [Author Response · Author response to Decision Letter 0]

29 Aug 2022

Thank you for reviewing the above referenced manuscript. The comments were very useful and the manuscript was amended accordingly. All changes made are highlighted in yellow in the revised version of the manuscript. Please see below the answers for all queries.

Do not hesitate to contact us if you require any further information.

Journal Requirements:

Comment 1: Please ensure that your manuscript meets PLOS ONE's style requirements, including those for file naming. 

Answer 1: We thoroughly revised instruction to authors and made all arrangements to the files and figures to meet PLOS ONE's style requirements. 

Comment 2: We note that the grant information you provided in the ‘Funding Information’ and ‘Financial Disclosure’ sections do not match. When you resubmit, please ensure that you provide the correct grant numbers for the awards you received for your study in the ‘Funding Information’ section.

Answer 2: As recommended, we provide the correct grant numbers for the awards in the ‘Funding Information’ section.

Reviewer #1

Thank you for your kind review. We incorporated your suggestions in the manuscript as follows:

Comment 1: Prediction scores as Wells and Geneva for pretest probability of PE validated in outpatient settings are occasionally used in the ICU. However, none of these clinical prediction rules have been validated in the ICU population. Therefore, more researches are needed to identify features that help to accurately identify patients with PE in this setting, and to develop a suitable prediction rule for critically ill patients with suspected PE. I think that in this study I do not see the relevance of considering these scores to analyze the usefulness of the multiorgan pocus. Please, comment about it.

Answer 1: This is an interesting point and we completely agree with the reviewer. The Wells and Geneva scores are used in an intensive care unit, but they are not validated in critically ill patients. There is no robust literature on the diagnostic investigation of PE in this population of patients. A previous study by our group (Ref 6, Wells and Geneva Scores Are Not Reliable Predictors of Pulmonary Embolism in Critically Ill Patients: A Retrospective Study) demonstrated that these scores were not good risk predictors for acute PE in critically ill patients, but the study was retrospective and we would like to confirm these findings in a prospective study. Then, we initially designed the study to include these scores in the analysis. However, our prospective results also showed low accuracy of these scores in critically ill patients with suspected pulmonary embolism (PE). Combining Wells and Genebra scores to mutiorgan ultrasound did not result I increased accuracy beyond the accuracy of multiorgan ultrasound. Therefore, we did not recommend to combine these scores with multiorgan ultrasound (please refer to Table 3 and Figure 1). We clarify this point in Discussion section (page 17, paragraph 1), as follows:

“Conversely, variables classically associated with PE, such as troponin and NT-proBNP [24], age [23], and the revised Geneva score [5], did not perform well in our population, in agreement with our previous retrospective results [6]. In this prospective study, combining the Wells and Geneva prediction scores with multiorgan scans did not increase diagnostic accuracy.”

Comment 2: In Table 2, define and explain better what it means “Alternative diagnosis on lung ultrasound”.

Answer 2: To better clarify what “Alternative diagnosis on lung ultrasound” means, we included a statement in Methods section (page 6, paragraph 2), as follows:

“The presence of one or more of the following lung abnormalities was considered an alternative diagnosis to PE on lung US: 1- the absence of pleural slip suggesting the presence of pneumothorax; 2- the presence of a hypoechoic pleural-based lesion suggesting consolidation or pulmonary atelectasis; 3- the presence of three or more B lines in an intercostal space in non-dependent lung areas suggesting alveolar-interstitial edema; 4- the presence of homogeneous anechoic area in a dependent lung area suggesting pleural effusion [15,16].”

Comment 3: Lung US performed by authors was based on diagnosis of acute respiratory failure method published on 2008. I would like to suggest that you make a more current and didactic classification shown in systematic review by Falster C et al. 

Answer 3: Thank you for this important note. We used the paper by Falster et al. (Ref 16, please refer to References) to define the item "differential diagnosis" in pulmonary ultrasound. This point was not clearly stated in our article, but it is now better clarified as presented in the answer above (comment and answer 2) and in Methods section (page 6, paragraph 2). Unfortunately, some relevant points from the meta-analysis by Falster et al. were not incorporated into our POCUS analysis, for instance, the identification of pulmonary infarctions and McConnell’s sign. Despite the high specificity of these two findings, their recognition would require long training and senior expertise, but we chose a more easily reproducible approach.

Reviewer #2 

This is an interesting study that looked into utilization of POCUS with clinical and lab values to diagnosed PE in comparison with CTPA.

Comment 1: "A noninvasive diagnostic strategy to help with timely decisions is important in critically ill patients" I think this statement should be modified as CTPA is already noninvasive, I would consider changing it.

Answer 1: We agree with the reviewer and changed the sentence as follows (page 4, paragraph 4):

“Therefore, alternative diagnostic strategies to help with timely decisions are important in critically ill patients.”

Comment 2: Definition of altered lung result on the basis of 3 or more altered fields may miss focal lung abnormalities like consolidation or atelectasis and this definition may under-diagnose alternative diagnoses to PE.

Answer 2: We agree with the reviewer. In all patients, we examined eight lung fields and collected two different variables in pulmonary ultrasound: 1- the presence of abnormal pulmonary findings; and 2- the presence of alternative pulmonary diagnosis (as better clarified in Methods section - page 6, paragraph 2). The absence of an alternative diagnosis on lung ultrasound (all lung fields considered) showed better performance and provided a more accurate information than the results of abnormal lung findings. Then we presented data on pulmonary ultrasound in a more objective way, referring only to the presence of alternative pulmonary diagnosis, which considered all lung fields examined. 

The presence or absence of differential diagnosis on pulmonary ultrasound showed better performance in our analysis, being included in our final model of better diagnostic accuracy (please refer to Table 2). The definitions of alternative pulmonary diagnosis used in the article are the following: 1 - absence of pleural slip suggesting the presence of pneumothorax; 2 - presence of a hypoechoic pleural-based lesion suggesting consolidation or atelectasis; 3 - presence of three or more B lines in an intercostal space in no dependent lung areas suggesting alveolar-interstitial edema; 4 - presence of homogeneous anechoic area in dependent lung area suggesting effusion". Among these items, consolidations and atelectasis might be present in only one lung field, but due to its clinical relevance was considered an alternative pulmonary diagnosis. In this sense, we do not believe we have missed or under-diagnosed alternative lung diagnosis to PE.

Comment 3: The authors did not specify if RV dysfunction was acute or chronic as presence of previous RV dysfunction may confound the findings significantly. Also, I believe all patients with previous RV dysfunction should have been excluded from the study.

Answer 3: This is a very important point to clarify. We consider all patients to have acute RV dysfunctions. Few patients had previous echocardiography, none of them with right ventricular dysfunction or pulmonary hypertension. We assumed that patients without previous echocardiography would not have right ventricular dysfunction or pulmonary hypertension. We included a statement in Results, to better clarify the topic, as follows (page 11, paragraph 1 ):

“Of the 37 patients with positive findings for PE on CTA, 22 (60%) had RV dysfunction on POCUS TTE, with 63% sensitivity and 85% specificity. All patients were considered to have acute RV dysfunction.”

Comment 4: There was no mention of patients with history of previous PE or being treated for PE as these also may confound the findings.

Answer 4: This is a very important point in the evaluation of patients with suspected PE. Information on previous history of PE or VTD was analyzed in all patients, as this is one of the items used to calculate the Wells score, which was performed for all patients. In our sample, only two patients had previous PE or VTD and none of them were on treatment for PE or DTV in the last six months. This information was included in Results section (page 9, paragraph 1 of the results):

“Eighty-eight critically ill patients were included in the study. Two patients had a history of previous PE or DVT, but none of them were being treated for PE or DVT in the last six months.”

Comment 5: Were the images acquired by the fellows saved and reviewed by more experienced physician to concur with accuracy of ultrasound image acquisition given that ultrasound is operator dependent?

Answer 5: When we design the paper, we had a specific concern about this point. Critical care fellows were specifically trained for ultrasound image acquisition for the protocol for three months before to start the inclusion of patients. All images were saved and revised by a senior physician (author R.B.A). This point is clarified in Material and Methods (page 6, paragraph 1), as follows:

“POCUS examinations were performed independently by two critical care fellows with advanced training in POCUS who had undergone three months of training in image acquisition specifically for the study protocol. All images were saved and reviewed by a senior physician if necessary.”

Comment 6: The authors mentioned specificity of 100% in the abstract and then mentioned specificity of 96% in the results section, this should be corrected and clarified.

Answer 6: Thanks for pointing this error. The correct value for specificity is 96%. The information was corrected accordingly in Abstract (page 3, paragraph 1):

“Combining these three findings resulted in an area under the curve of 0.85 (95% CI, 0.77-0.94), with 50% sensitivity and 96% specificity.”

Comment 7: Given sensitivity of 50%, I think it is really difficult to state in the conclusion the diagnostic accuracy in high at this point. I would be more cautious with such a statement as the data presented does not fully support this conclusion.

Answer 7: We agree with the reviewer. The statement was changed as follows (page 18, paragraph 1, and in Abstract, page 3):

In Conclusion section:

“In conclusion, multiorgan POCUS combined with laboratory data has acceptable diagnostic accuracy for PE compared with CTA and might be helpful in the ICU setting, providing an alternative method when performing a CTA is too risky and reducing unnecessary tests when the probability of disease is too high.”

In Abstract section:

“Multiorgan POCUS combined with laboratory data has acceptable diagnostic accuracy for PE compared with CTA. The combined use of these methods might reduce CTA overuse in critically ill patients.”

Reviewer #3 

Comment 1: I had a very difficult time determining what proportion of PEs were sub-segmental in nature. 

Answer 1: As stated in Material and Methods (page 8, paragraph 1), thrombus location was defined according to the caliber of the affected vessel and classified as follows: main artery, lobar, segmental, or subsegmental. Scans were considered negative in the presence of adequate opacification of the pulmonary artery without filling defects, according to Moores et al. (included Ref 17, please refer to References). We had four patients with subsegmental PE (11% of patients with PE). This information was added to Results (page 9, paragraph 2), as follows:

“Of 88 patients, 37 (42%) had PE detected on pulmonary CTA examination. Of these, 12 patients (32%) had PE in the main artery, three (8%) in the lobar branch, 18 (49%) in the segmental branch, and four patients (11%) in the subsegmental branch.”

Comment 2: The paper also requires significant language editing.

Answer 2: Thanks for this important contribution to the final quality of the manuscript. A new round of language editing was performed. Paraphrasing that did not change the meaning of sentences is not highlighted in yellow. (Attached certificate in "response to reviewers" file)

Comment 3: Abstract is lacking key information i.e. the sample size included in the study., in order to provide in the abstract an informative and balanced summary of what was done and what was found in the study, in accordance with STROBE statement.

Answer 3: Abstract was fully revised and all key information (including sample size) was amended accordingly with STROBE statement, in order to provide in the abstract an informative and balanced summary of what was done and what was found in the study. Please refer to Abstract section (page 2).

---

## [Decision Letter · Decision Letter 1]

19 Sep 2022

PONE-D-22-18103R1Diagnostic accuracy of multiorgan point-of-care ultrasound compared with pulmonary computed tomographic angiogram in critically ill patients with suspected pulmonary embolism.PLOS ONE

Dear Dr. Girardi, Adriana

Thank you for submitting your manuscript to PLOS ONE. After careful consideration, we feel that it has merit but does not fully meet PLOS ONE’s publication criteria as it currently stands. Therefore, we invite you to submit a revised version of the manuscript that addresses the points raised during the review process.

ACADEMIC EDITOR:Please, note the reviewer 2 request and answer it. My decision is justified on PLOS ONE’s publication criteria and not, for example, on novelty or perceived impact.

We look forward to receiving your revised manuscript.

Kind regards,

Patricia R Prado

Academic Editor

PLOS ONE

Journal Requirements:

Reviewers' comments:

Reviewer's Responses to Questions

**Comments to the Author**

1. If the authors have adequately addressed your comments raised in a previous round of review and you feel that this manuscript is now acceptable for publication, you may indicate that here to bypass the “Comments to the Author” section, enter your conflict of interest statement in the “Confidential to Editor” section, and submit your "Accept" recommendation.

Reviewer #2: All comments have been addressed

Reviewer #3: All comments have been addressed

2. Is the manuscript technically sound, and do the data support the conclusions?

Reviewer #2: Yes

Reviewer #3: Yes

3. Has the statistical analysis been performed appropriately and rigorously? 

Reviewer #2: -

Reviewer #3: Yes

4. Have the authors made all data underlying the findings in their manuscript fully available?

Reviewer #2: Yes

Reviewer #3: Yes

5. Is the manuscript presented in an intelligible fashion and written in standard English?

Reviewer #2: Yes

Reviewer #3: Yes

6. Review Comments to the Author

Reviewer #2: I would like to thank the authors for addressing the comments on the manuscript. I think it would be appropriate that the authors acknowledge the lack of exclusion of previous RV dysfunction as weakness in the discussion section.

Reviewer #3: I am quite satisfied with the author's revisions of this manuscript. In particular, the attention paid to the inclusion of the STROBE statement elevates the project considerably.

7. PLOS authors have the option to publish the peer review history of their article (what does this mean?). If published, this will include your full peer review and any attached files.

Reviewer #2: No

Reviewer #3: **Yes: **E.L.

---

## [Author Response · Author response to Decision Letter 1]

29 Sep 2022

Dear reviewers, we thank you for revising our manuscript. We make changes as requested. Answer follows below:

Thank you for reviewing the above referenced manuscript. All changes made are highlighted in yellow in the revised version of the manuscript. Please see below the answers for all queries.

Journal Requirements:

Comment 1: Please review your reference list to ensure that it is complete and correct. If you have cited papers that have been retracted, please include the rationale for doing so in the manuscript text, or remove these references and replace them with relevant current references. Any changes to the reference list should be mentioned in the rebuttal letter that accompanies your revised manuscript. If you need to cite a retracted article, indicate the article’s retracted status in the References list and also include a citation and full reference for the retraction notice.

Answer 1: We thoroughly revised the reference list. In our last version of the manuscript, we added reference 17, which defines findings suggestive of PE on pulmonary computed tomographic angiogram. Also, we modified the position of reference 16 (previously reference 23), mentioned now in Materials and Methods and Discussion sections. In the current version of the manuscript, the references are all complete and correct. No reference cited that have been retracted. Reference list meets PLOS ONE style requirements.

Reviewer #2 

I would like to thank the authors for addressing the comments on the manuscript.

Comment 1: I think it would be appropriate that the authors acknowledge the lack of exclusion of previous RV dysfunction as weakness in the discussion section.

Answer 1: A statement was added to the Discussion section as follows (page 18, paragraph 1):

“Third, a formal exclusion of patients with previous RV dysfunction was not performed. However, no patients were suspected to have previous pulmonary hypertension based on previous echocardiography data and clinical data. Fourth, this is a single-center study.” 

Reviewer #3

Comment 1: I am quite satisfied with the author's revisions of this manuscript. In particular, the attention paid to the inclusion of the STROBE statement elevates the project considerably.

Answer 1: We thank the reviewer for spending his time to improve your manuscript.

---

## [Editor Report · Decision Letter 2]

2 Oct 2022

Diagnostic accuracy of multiorgan point-of-care ultrasound compared with pulmonary computed tomographic angiogram in critically ill patients with suspected pulmonary embolism.

PONE-D-22-18103R2

Dear Dr. Girardi Adriana

We’re pleased to inform you that your manuscript has been judged scientifically suitable for publication and will be formally** accepted **for publication once it meets all outstanding technical requirements.

Kind regards,

Patricia Rezende do Prado

Academic Editor

PLOS ONE

---

## [Editor Report · Acceptance letter]

10 Oct 2022

PONE-D-22-18103R2 

Diagnostic accuracy of multiorgan point-of-care ultrasound compared with pulmonary computed tomographic angiogram in critically ill patients with suspected pulmonary embolism. 

Dear Dr. Girardi:

I'm pleased to inform you that your manuscript has been deemed suitable for publication in PLOS ONE. Congratulations! Your manuscript is now with our production department. 

Kind regards, 

on behalf of

Dr. Patricia Rezende do Prado 

Academic Editor

PLOS ONE